# Are Inositol Polyphosphates the Missing Link in Dynamic Cullin RING Ligase Regulation by the COP9 Signalosome?

**DOI:** 10.3390/biom9080349

**Published:** 2019-08-07

**Authors:** Xiaozhe Zhang, Feng Rao

**Affiliations:** Department of Biology, Institute of Neuroscience, Guangdong Provincial Key Laboratory of Cell Microenvironment and Disease Research, Southern University of Science and Technology, Shenzhen 518055, China

**Keywords:** Cullin ring ubiquitin ligases (CRL), neddylation, COP9 signalosome, CSN, deneddylase, inositol hexakisphosphate (IP6), inositol pyrophosphates, IP6K

## Abstract

The E3 ligase activity of Cullin RING Ligases (CRLs) is controlled by cycles of neddylation/deneddylation and intimately regulated by the deneddylase COP9 Signalosome (CSN), one of the proteasome lid-CSN-initiation factor 3 (PCI) domain-containing “Zomes” complex. Besides catalyzing the removal of stimulatory Cullin neddylation, CSN also provides a docking platform for other proteins that might play a role in regulating CRLs, notably protein kinases and deubiquitinases. During the CRL activity cycle, CRL–CSN complexes are dynamically assembled and disassembled. Mechanisms underlying complex dynamics remain incompletely understood. Recently, the inositol polyphosphate metabolites (IP6, IP7) and their metabolic enzymes (IP5K, IP6K) have been discovered to participate in CRL–CSN complex formation as well as stimulus-dependent dissociation. Here we discuss these mechanistic insights in light of recent advances in elucidating structural basis of CRL–CSN complexes.

## 1. Introduction

The “Zomes” complexes refer to three proteastasis-related machineries with subunits containing the PCI (proteasome lid-CSN-initiation factor 3) domain and the MPN (Mpr1/Pad1 N-terminal) domain: the translation initiation factor-3, the 19S regulatory subunit of the 26S proteasome, and the COP9 singalosome (CSN). The CSN is a 450-kDa multi-protein complex conserved from fungi to plants and humans [1,2]. CSN was first identified and purified from *Arabidopsis* and named COP9 because mutations in its subunits lead to Constitutive Photomorphogenesis (COP), even when seedlings were grown in the dark [3,4]. Concurrently, in an attempt to identify 26S proteasome components, a separate group a separate purified human CSN as an eight-subunit complex called signalosome (Sgn) [5]. The name COP9 signalosome, abbreviated as CSN, was eventually adopted to incorporate both the plant and human name, with the eight CSN subunits named CSN1 to CSN8 based on descending molecular weight. CSN mediates many fundamental and disease aspects of life, including, among others, cell cycle control, DNA repair, stemness, development, cancer and cardiovascular diseases [6,7,8,9,10].

Although complex-independent functions have been reported for certain CSN subunits [11,12,13], the primary biochemical role of CSN is to regulate the Cullin RING ubiquitin ligases (CRLs) as a deneddylase holoenzyme. CRLs are composed of four modules: a Cullin scaffold protein (Cul1-3, 4A/B, 5 or 7), an E2-interacting RING protein (Rbx1/2, also called Roc1/2) that heterodimerizes with the Cullins, a Cullin-specific adaptor protein, and a substrate receptor module that directly recognize ubiquitylation targets [14,15]. As part of a three enzyme cascade (E1-E2-E3) that together ubiquitylates protein substrates for proteasomal degradation or for signaling, CRLs are the largest family of E3s, targeting approximately 20% of the proteome’s turnover [16]. The numerous CRL substrates are involved in nearly every aspect of biological processes, including cell cycle progression, DNA repair, metabolism, autophagy, development, immunity, cancer, and so on. Accordingly, CRLs may be targeted for therapeutic intervention when specific CRL substrates become the cause of a disease, most prominently in cancer research [17,18]. Recently, the rise of technologies such as proteolysis targeting chimera (PROTAC), which tailors CRLs to degrade specific proteins of therapeutic relevance via the assistance of intermolecular “glue”-like small molecules, has empowered CRLs as cutting-edge drug discovery tools [19].

The E3 ligase activity of CRLs is controlled by reversible Cullin neddylation (modification by the ubiquitin-like protein Nedd8) and deneddylation [15]. Neddylation, catalyzed by an E1-E2-E3 enzyme cascade analogous to the ubiquitylation machineries, enhances CRL activity by inducing productive E2 engagement with Rbx1/Roc1 [20,21]. The COP9 Signalosome is the deneddylase targeting all neddylated Cullins [22,23]. In vitro, CSN inhibits CRL activity by catalyzing Nedd8 removal, and by binding and sequestering deneddylated CRL [23,24,25]. However, genetic disruption of CSN subunits can lead to constitutive CRL activation and self-destruction, eventually diminishing CRL function [26,27]. This apparent inconsistency, known as the “CSN paradox”, indicates that CSN protects CRL from being aberrantly active under basal condition, while enabling proper CRL dissociation and activation in a signal-dependent manner [28]. Thus, the assembly and disassembly of CRL–CSN complexes is at the center stage of CRL regulation.

Recent progress in structural studies of the CRL–CSN complex has revealed key mechanistic insights on CRL–CSN assembly interface, which is reviewed here in conjunction of our work suggesting that the inositol polyphosphate metabolic pathway might provide the missing link in dynamic Cullin RING ligase regulation by the COP9 signalosome.

## 2. Structural Basis of CRL–CSN Supercomplexes

Structural features of CRL have been elucidated by early crystallographic studies (Figure 1A) [21,29,30]. Briefly, the N-terminal helical repeat domain of Cullins recruits adaptors, which in turn recruits substrate receptors; whereas the C-terminal domain of Cullins (Cullin^CTD^), comprised of a four-helix bundle (4HB), an α/β and two winged-helix (WHA and WHB) subdomains, heterodimerizes with Rbx1 via its α/β subdomain and recruits E2 ubiquitin conjugating enzymes primarily via Rbx1. In this way, the Cullin/Rbx1 serves as a scaffold to bring substrate and ubiquitin-charged E2 in proximity for transfer to take place. This modular feature applies to all CRLs despite their distinct adaptor and substrate receptors. Such commonality also suggests that the regulatory mechanisms of CSN are generalizable, as discussed below.

Crystal structure of the CSN holo-complex is only recently solved by Thoma and associates [31] (Figure 1B). The holo-complex structures verified the previously reported structures of CSN subunits [33,34,35,36]. Moreover, they clearly reveal that the overall structural composition and enzymatic nature of the CSN complex bare resemblance to the other two ZOMES complexes: the lid complex of the 19S regulatory particle of the 26S proteasome and the eukaryotic initiation factor 3 complex [37]. All three complexes regulate protein homeostasis and are composed of subunits with MPN zinc-metalloprotease domain for cleaving isopeptidase bonds, and PCI domain for oligomerization, suggesting that they are evolutionarily related.

Six CSN subunits, CSN1, CSN2, CSN3, CSN4, CSN7, and CSN8, contain a C-terminal PCI domain, the edge-to-edge interactions of which provide the driving force for oligomerization. Additionally, the carboxy-terminal α-helices of all eight CSN subunits form a helical bundle to uphold the integrity of the eight-subunit complex. On top of the helical bundle, the two non-PCI subunits, CSN5 and CSN6, form a dimmer via their MPN domains. CSN5 is the catalytic subunit whose MPN domain has conserved metalloprotease active site, whereas that of CSN6 is degenerate and catalytically incompetent. Interestingly, while the catalytic mechanism of CSN is analogous to other MPN metalloprotease, CSN is catalytically inactive in this structure and also in solution when assayed using ubiquitin-rhodamine as an artificial deubiquitinase substrate. This is because a CSN5 loop containing Glu104 occludes the active site with Glu104 interacting with the catalytic Zn^2+^ ion. Mutating Glu104 enhances the isopeptidase activity against ubiquitin-rhodamine, supportting the conclusion that CSN is normally in an autoinhibited state and requires an activation mechanism to deneddylate CRLs.

Several recently obtained EM structures of CRL–CSN complexes shed light on mechanisms underlying the formation of CRL–CSN supercomplexes as well as CRL-induced CSN activation (Figure 2). The first report of molecular models of CSN in complex with two CRL1/SCF subcomplexes (SCF^Skp2/Cks1^ and SCF^Fbw7^) was obtained at 20 Å resolution by negative-staining electron microscopy (EM) [24]. Several observations made from this low-resolution EM map were subsequently verified by cryo-EM maps of CRL4A-CSN and SCF-CSN complexes obtained at higher resolution (CSN–_N8_CRL4A at 6.4 Å [38] and CSN–_N8_SCF^Skp2/Cks1^ at 7.2 Å [39]). As such, they are discussed altogether here.

Regarding CRL–CSN supercomplex assembly mechanisms, all EM maps suggest that CSN binds to CRL at multiple sites, with the most extensive contact being CSN2 embracing the C-terminal portion of Cul1/Rbx1 (Cul1^CTD^/Rbx1). Two other points of contact include the sandwich of Rbx1 RING domain by CSN2 and CSN4, and that CSN1 and CSN3 are in the vicinity and likely interact with SCF adaptor/substrate receptor pair Skp1/Skp2/Cks1 or the CRL4A adaptor DDB1. Overall, these structures agreed with each other in outlining major CRL–CSN contact sites, although higher resolution structures are still required to visualize atomic details of the CRL–CSN interface.

The elucidated mechanistic insights on CRL–CSN binding interface help explain several biochemical and enzymatic observations. First, CSN2 embracing of Cul1^CTD^/Rbx1 and the sandwich of Rbx1 RING domain by CSN2 and CSN4 would prevent SCF from interacting with the E2 enzyme CDC34, which is recruited to SCF via Rbx1 RING domain and a basic canyon at Cul1^CTD^ [40]. Consistent with such competition, CSN strongly inhibits CRL4^DDB2^ autoubiquitylation or SCF substrate uniquitylation in a substrate peptide ubiquitylation assay [24,41]. Second, CSN binding could also obstruct the neddylation E2 in a similar manner, which is consistent with the inhibition of CRL neddylation by catalytically inactive CSN [24]. Third, the contact between CSN1/CSN3 and adaptor and/or substrate receptor are incompatible with substrate access. Consistently, CSN addition prevented p-p27 binding to SCF^Skp2/Cks1^. In reciprocity, the presence of CRL substrates [24,25,42] or a substrate receptor binding DNA fragment [38], markedly inhibits Cullin deneddylation, conceivably by interfering with CSN–CRL complex formation. Of note, the mutually exclusive binding of substrate or CSN could be an important determinant in CRL–CSN complex dissociation, as discussed below.

In addition to identifying CRL–CSN contact interfaces, the CSN–_N8_CRL4A and CSN–_N8_SCF^Skp2/Cks1^ structures further reveal conformational rearrangements possibly leading to CSN activation. To fit the electron density map of the CRL–CSN complexes, several subunits or their subdomains, specifically the N-terminal portions of CSN2 and CSN4, as well as the MPN-domains of CSN5 and CSN6, the RING domain of Rbx1, the WHB domain of Cullins, and Nedd8, undergo significant conformational movement. Two driving forces underlie these conformational changes. First, the N-terminal portions of CSN2 and CSN4 interact with the RING domain of Rbx1, and the WHB domain of Cullins, conformation rearrangement in this region is consistent with an induced-fit mechanism to strengthen cognate enzyme-substrate recognition; Second, because the CSN5 MPN metalloprotease active site is self-obstructed in the absence of neddylated CRL, conformational rearrangement in this region is necessary for CSN activation upon Nedd8 recognition. Based on these observations and rationales, it was proposed that binding to a neddylated CRL induces movement in CSN4, which would be expected to alter the CSN4–CSN6 interacting interface and eventually rearrange the CSN5–CSN6 dimer to activate CSN5. Whether and how conformational changes at the N-terminal segments of CSN2 and CSN4 are truly transduced to the catalytic site still require further structural validations. But mutational data support that this process likely involves the CSN4–CSN6 interface [38,39]. Regardless, stringent requirement of CRL-binding to activate CSN and mutual structural rearrangement during the process suggests that CSN is dedicated to CRL deneddylation. Indeed, neddylated Cullins are the only known CRL substrate.

Apart from revealing CRL–CSN contact interfaces and potential CSN activation mechanisms, one key insight gained from the cryo-EM structures of CRL–CSN complexes is the steric clash between CSN and incoming ubiquitylation substrates [24,25,41]. This structural feature has two implications. First, steric incompatibility between CSN and substrate indicates that only substrate-free CRL can be inactivated by CSN. This then points towards a dedicated time window for CSN to function during the CRL activity cycle, i.e., after ubiquitylated substrates have been extracted from CRL–E2 complex, likely via a p97 dependent pathway [43]. Second, new incoming substrate could provide a driving force to dissociate inert CRL–CSN complexes by sterically dislodging CSN, thereby initiating a new round of Cullin neddylation and substrate ubiquitylation. In the case of CRL4^DDB2^, a damage-mimicking DNA duplex that binds avidly to DDB2 can indeed displace CSN [23]. However, it remains to be demonstrated if an ubquitylated substrate can also dissociate pre-formed CRL–CSN complexes, especially given that CSN binds to deneddylated CRL with decent, nanomolar-level affinity [39].

The substrate-induced CSN dissociation model suggests that CSN-dissociated CRL are always substrate-bound. This then brings a dilemma regarding the action of CAND1 (Cullin-associated Nedd8-dissociated protein 1), which wraps around non-neddylated CRLs to exchange substrate receptor [44,45,46]. Does CAND1 exchange substrate receptors with bound substrates? If so, this would make substrate-binding futile, and is also inconsistent with the actual function of CAND1, i.e., to exchange substrate receptor such that available substrates are targeted for degradation.

An alternative possibility is that the CRL–CSN complexes are dynamically assembled into high- and low-affinity states, with the low-affinity state prone to dissociation or subjected to CAND1 competition (Figure 3). Differential affinity states of CRL–CSN complexes could be determined by post-translational modifications on CSN and/or an extra factor mediating CRL–CSN interactions. In this regard, the recent identification of inositol polyphosphates and their metabolic enzymes as emerging modulators of CRL–CSN complex formation offer a new perspective on complex dynamics, as detailed below.

## 3. The Role of Inositol Polyphosphates and their Synthases in CRL–CSN Complex Dynamics

The higher inositol polyphosphates originate from the second messenger inositol triphosphate (IP_3_), which is produced from the hydrolysis of PIP2 by PLC, while PLC is activated by G_q_ protein-coupled receptors (GPCRs) on cell membrane [47]. IP3 is then phosphorylated by a series of inositol phosphate kinases (IPKs), including inositol 1,4,5-trisphosphate 3-kinase (IP3K), inositol 1,3,4-triphosphate 5/6-kinase (ITPK1), inositol polyphosphate multikinase (IPMK), and inositol 1,3,4,5,6-pentakisphosphate 2-kinase (IPPK/IP5K), to generate IP4 to IP6 [48]. IP6 is the most abundant inositol phosphate, but a portion of it can be further phosphorylated to generate the inositol pyrophosphates IP7 and IP8, which are unstable molecules containing energetic pyrophosphate bond(s) and are believe to play signaling roles [49]. These higher inositol polyphosphates are highly conserved from yeast to human, but are less characterized compared to IP3.

In purifying ITPK1 from a calf brain, Majerus and associates identified a copurifying eight-protein complex with components ranging in size from 60 to 20 kDa, which turns out to be CSN based on Mass-Spec [50]. ITPK1 directly interacts with CSN1 [51]. Because ITPK1 purified this way can phosphorylate c-Jun, it was proposed that ITPK1 is the CSN-associated kinase that is known to phosphorylate c-Jun. However, PKD and CK2 are the true CSN-associated protein kinase [52]. Whether recombinant ITPK1 directly phosphorylate proteins was not verified. As such, the physiological significance of CSN association with an inositol phosphate kinase remained unclear for some while.

Intriguingly, Zheng and associates later found IP6 to be located in a plant CRL1 complex [53]. Specifically, IP6 is near the center of the TIR1–LRR fold in close proximity to auxin binding site. TIR1 is a plant auxin receptor and a F-box protein that bind to SCF complex to ubiquitin substrates. Auxin binding to TIR1 can promote the interaction between TIR1 and Aux/IAA protein, activate SCF^TIR1^ complex to ubiquitinate Aux/IAA protein for gene transcription [51,52]. In the crystal structure of TIR1 and ASK1, which is the adaptor of SCF^TIR1^, IP6 closely interacts with TIR1, via ten positively charged residues. Thus, IP6 is also an auxin receptor co-factor, and mediates auxin-dependent substrate ubiquitylation. Whether IP6 also targets other SCF or CRL complexes is unknown.

More recently, Rao et al. reported that inositol hexakisphosphate kinase-1 (IP6K1), another IPK that phosphorylates IP6 to IP7, interacts with CRL4 via direct binding to the CRL4 adaptor DDB1 [54]. IP6K1 is not degraded by CRL4. Rather, co-expression suppresses the increase in total cellular ubiquitylation levels caused by Cul4A-Rbx1-DDB1 overexpression, suggests that IP6K1 regulates cellular CRL4 activity. Conversely, increasing amount of DDB1 suppresses IP6K1 activity in vitro, suggesting that the IP6K1–CRL4 complex is inactive. Consistently, IP6K1 promotes CSN binding to and sequestration of CRL4, thereby forming an inert IP6K1–CRL4-CSN ternary complex under basal condition. Interestingly, UV radiation, a stimuli known to dissociate CSN–CRL4 for DNA repair [55], also dissociates IP6K1 from CRL4–CSN, via yet unclear mechanisms. Nonetheless, UV upregulates cellular IP7 levels, whereas kinase-dead IP6K1 mutant, or the IP6K-specific inhibitor TNP, can attenuate UV-induced CRL4–CSN complex disassembly, suggesting that IP6K1-dependent IP6 to IP7 conversion promotes CRL4–CSN dissociation.

To understand the roles of IP6 and IP7 in CRL–CSN complex dynamics, biochemical reconstitution of CRL–CSN interactions using purified recombinant proteins was performed, revealing two novel mechanisms of CRL–CSN interaction not identified by the cryo-EM studies [56]. First, electrostatic interaction between the N-terminal acidic tail of CSN2 and the conserved C-terminal basic canyon of Cullins is identified. This mechanism of interaction is reminiscent of that between Cullin’s basic canyon and the C-terminal acidic tail of the E2 CDC34 [40,57], thus providing the biochemical explanation for the earlier observation that CSN competes with CDC34 to inhibit CRL [24]. Second, IP6 directly stimulates the in-vitro binding between Cul4A/Roc1 and CSN2 at nanomolar concentrations, whereas the IP6 synthase IP5K interact with the CRL–CSN complex, and mediates the stability, neddylation and activation of cellular CRL1 and CRL4 [43]. Importantly, although IP7 can also promote CRL4A–CSN2 binding in vitro, it does so with threefold lower potency than IP6 [43]. Together with the generally higher abundance of IP6 (20 times of IP7) [58], these data suggest a working model whereby IP6K1-catalyzed IP6 conversion to IP7 could potentially increase the percentage of lower-affinity CRL4-CSN complexes, thereby rendering CRL–CSN complexes amenable to dynamic dissociation or to competition by CAND1 or ubiquitylation substrates (Figure 3).

These findings raise the tantalizing possibility that IP_6_ and its metabolic enzymes are messengers receiving upstream signals to control CRL–CSN complex dynamics and CRL activity. However, some outstanding questions remain. First, how general is the mechanism? Is CRL regulation by IP6/IP7 dynamics is generally applicable for all CRL4 ligases or only a subset with specific substrate receptors, and whether the same principle applies to other Cullin members? Second, what is the biochemical and structural basis for differential regulation of CRL–CSN interactions by IP6 and IP7? Is IP6 acting as an intermolecular “glue” at the interface between CRL and CSN? Could a higher resolution Cryo-EM structure of CRL–CSN complex reveal the presence of a IP6/IP7 molecule at the interface? Third, How does IP7 differ from IP6? Measuring the K_on_ and K_off_ between CRL and CSN in the presence of IP6 or IP7 would be essential to validate the working model that IP7-bridged CRL–CSN complex are more prone to dissociation than that coordinated by IP6. Fourth, how does IP6 influence the competition between CSN and E2 CDC34? Is IP6 a decisive factor in CSN-dependent E2 release from CRL? Fifth, thus far, three IPKs, ITK1, IP5K and IP6K1, have been found to interact with CRL–CSN with apparently different direct binding partners. ITPK1 binds to CSN1 [51], whereas IP5K and IP6K1 bind to Cullin and the Cul4 adaptor DDB1, respectively [54,56]. Enzyme clustering is a characteristic property of the classical metabolic systems [59]. Might IPKs assemble as an enzymatic chain and efficiently channel intermediates for local production of IP6/IP7? Last but not the least, what is the molecular basis for UV-induced IP6K1–DDB1 dissociation and IP6K1 activation? Would other stimuli such as cell cycle transition involve a similar mechanism to activate CRL?

Although more questions are to be answered, the above findings bare translational values. Cullins neddylation is targetable. MLN4924, now known as Pevonedistat, is a small molecular inhibitor of the neddylation E1 (NAE), can prevent the neddylation of Cullins via inhibiting NEDD8 binding to NAE [16]. In cancer, Cullins are activated by augmented neddylation to degrade tumor suppressors [60]. MLN4924 can inhibit this process to prevent tumor growth [16]. In fact, Pevonedistat/MLN4924 is currently under phase III clinical trial as an anti-cancer agent (NCT03268954) [61]. IP6 can promote the interaction between Cul4A and CSN2 to enhance Cul4A deneddylation [43], so IP6 or IP6K1 could be potential cancer treatment targets.

## 4. Conclusions and Future Perspectives

The past few years has witnessed an explosion of scientific advance in understanding the structural mechanisms and physiologic implications of CRL regulation by CSN. Here we mainly focused on the small inositol polyphosphate metabolites, but other protein factors have also been uncovered, such as CSNAP [62], Glomulin [63], and Rig-G [64]. Further research is required to delineate the spatiotemporal mechanism of action of these new players. Such insights could contribute to our understanding on the important question of how CRL–CSN complexes are dynamically assembled and disassembled during the CRL activity cycle, despite their high binding affinity.

## Figures and Tables

**Figure 1 biomolecules-09-00349-f001:**
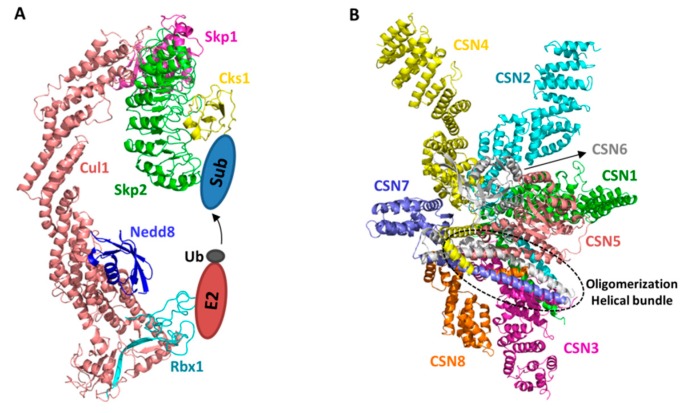
Structural models of a CRL (SCF ^Skp2/Cks1^) and the COP9 signalosome (CSN). Individual subunits are colored differently. The E2 and substrate locations are shown in the scheme. The helical bundle formed by the c-terminal helix of all eight CSN subunits is highlighted in circle (**A**). The SCF ^Skp2/Cks1^ complex is built as previously described [24], based on the crystal structures of SCF^Skp2^ [29], Skp1-Skp2/Cks1 [31], and Nedd8-Cul1^1-690^ [21]. The CSN model is a representation of a previous published crystal structure (PDB id: 4D10) [32] (**B**).

**Figure 2 biomolecules-09-00349-f002:**
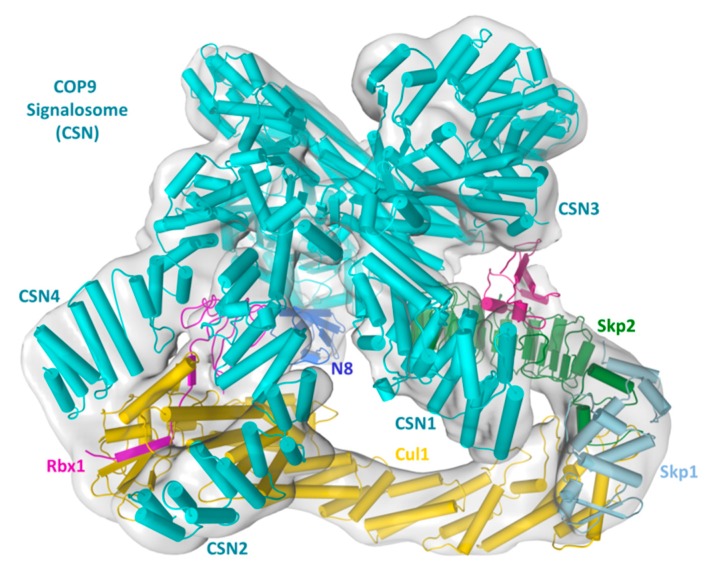
Fitting of the CSN-_N8_SCF^Skp2/Cks1^- electron density map (EMD-3401, 7.2Å) [39] with crystal structures of the subcomponents, followed by refinement with Molecular Dynamic Flexible Fitting (MDFF). Protein secondary structures are shown in pipes.

**Figure 3 biomolecules-09-00349-f003:**
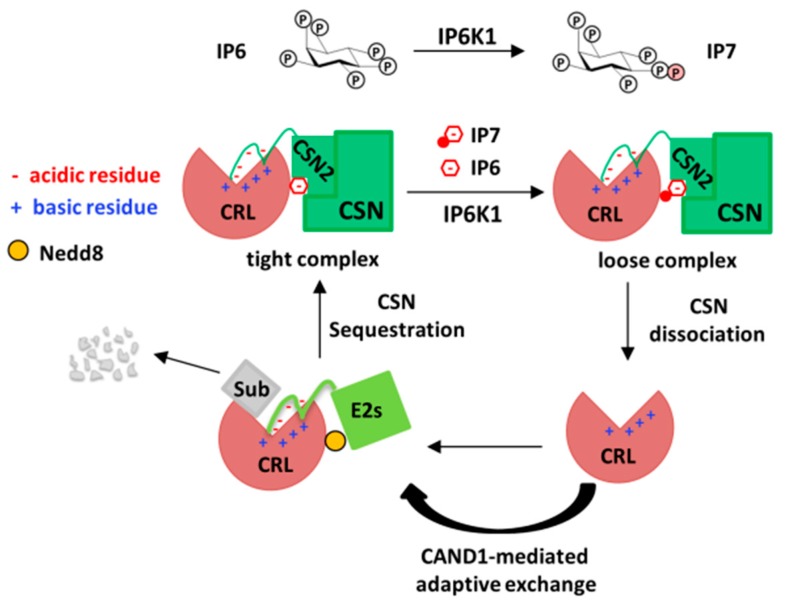
Scheme depicting the role of inositol polyphosphates in the assembly and disassembly of CRL–CSN complexes. Driven by the kinase IP6K1, the CRL–CSN complexes inter-converts between the IP_6_-bridged, high-affinity state and the IP_7_-brdiged, lower-affinity state, with the later prone to dissociation, permitting CAND1-mediated exchange of substrate receptors.

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
