# Peer review of "Are Inositol Polyphosphates the Missing Link in Dynamic Cullin RING Ligase Regulation by the COP9 Signalosome?"

_biomolecules, 2019, doi:10.3390/biom9080349_

Round 1

Reviewer 1 Report

This is a nice summary of the research advance on CNS-CRL interaction, and the potential role of inositol polyphosphates in this process. The role of inositol metabolism in ubiquitin proteasome regulation is highly interesting. A few minor comments for consideration.

1.     In Arabidopsis, IP6 has been identified as a co-factor for the auxin sensing by the SCF(Tir1) E3 ligase. Authors may want to visit this research topic (e.g., a nice review on this topic, https://www.nature.com/articles/nature05731), and determine whether related information should be added to this perspective.

2.     Line 154, it is not clear what the “GPCR 2nd messenger” means? Which GPCR?

3.     Whether the structure images have the permission to be published. It may be necessary to add a note in the figure legend to state that the images are modified from the original publications (citation).

4.     A few typos:

Line 28, “drak” should be “dark”

Line 60, remove one “that”

Line 97, “substrate”

Author Response

We are naturally pleased that this reviewer found our manuscript to be of general interest. Below please find our point by point response to the referee:

1.     In Arabidopsis, IP6 has been identified as a co-factor for the auxin sensing by the SCF(Tir1) E3 ligase. Authors may want to visit this research topic (e.g., a nice review on this topic, https://www.nature.com/articles/nature05731), and determine whether related information should be added to this perspective.

We thank the referee for pointing out this interesting work from the Ning Zheng lab, which actually provide the first link between IP6 and CRL. We have now discussed this work on page 8, line 225-234.

 2.     Line 154, it is not clear what the “GPCR 2nd messenger” means? Which GPCR?

IP3 is a generic 2nd messenger for Gq-coupled GPCRs. For clarity purposes we have rephrased the paragraph discussign the origin of inositol polyphosphates on page 7, line 204-206. 

3.     Whether the structure images have the permission to be published. It may be necessary to add a note in the figure legend to state that the images are modified from the original publications (citation).

We thank this referee again for alerting us to cite the original literature reporting the structures. These figures are not copied from original literature, but generated fresh based on electron density map published in online database (PDB or EMB). Where appropriate, we have added citation to the original literature, and also described how the figures are generated. (Line 463-469, and line 471-474).

4.     A few typos:

Line 28, “drak” should be “dark”

Line 60, remove one “that”

Line 97, “substrate”

We have corrected the typos. 

Thank you.

Reviewer 2 Report

The manuscript by Zhang and Rao satisfactorily summarizes the latest findings, obtained through structural studies, about the mechanisms that govern assembly of the CSN complex and CRL E3 ubiquitin ligases and the impact that substrates and inositol polyphosphates exert on the dynamics of such assembly/disassembly cycles.

The manuscript is concise yet complete, well written and clear. However, the non-expert might get lost while trying to visualize all dynamic interactions between CSN and CRLs described in the manuscript. Thus, the manuscript would benefit if a simple scheme summarizing the interplay between CSN, CRL and target-loaded substrate adapters for supercomplex assembly and disassembly is included. Such scheme could be similar in style to that shown in Figure 3.

Minor:

Line 154. The meaning of GPCR should be indicated. 

Several typos can be found in the manuscript. Some examples are:

Line 28: it is written drak instead of dark

Line 38: proteasombal instead of proteasomal

Line 39: singaling instead of signaling

Line 78: reveal instead of reveals

Line 92: Cu1 instead of Cul1

Line 97: ubstrate instead of substrate

Line 112: completion instead of competition

Line 126: mediates instead of mediate

Line: 154: 2nd instead of second

Line 195: principleapply instead of principle apply

Line 212: Gloulin instead of Glomulin

Line 219. Cyro-EM instead of cryo-EM

Author Response

We thank this referee for finding our manuscript in principle satisfactory. Below please find our point to point reply to the referee.

"The manuscript is concise yet complete, well written and clear. However, the non-expert might get lost while trying to visualize all dynamic interactions between CSN and CRLs described in the manuscript. Thus, the manuscript would benefit if a simple scheme summarizing the interplay between CSN, CRL and target-loaded substrate adapters for supercomplex assembly and disassembly is included. Such scheme could be similar in style to that shown in Figure 3."

We concur with this reviewer that a scheme summarizing the interplay between CSN, CRL and target-loaded substrate adapters for supercomplex assembly and disassembly would be help readers to understand the CRL activity cycle better. We have therefore re-drawn figure 3 of the manuscript to include a whole catalytic cycle.

Minor:

Line 154. The meaning of GPCR should be indicated. 

We have now indicated the meaning of GPCR in greater detail on page 7, line 204-206.

Several typos can be found in the manuscript. Some examples are:

Line 28: it is written drak instead of dark

Line 38: proteasombal instead of proteasomal

Line 39: singaling instead of signaling

Line 78: reveal instead of reveals

Line 92: Cu1 instead of Cul1

Line 97: ubstrate instead of substrate

Line 112: completion instead of competition

Line 126: mediates instead of mediate

Line: 154: 2nd instead of second

Line 195: principleapply instead of principle apply

Line 212: Gloulin instead of Glomulin

Line 219. Cyro-EM instead of cryo-EM

We thank this referee for carefully pointing our the typos, all of which have now been corrected.